# Mothers' health care seeking behavior for neonatal danger sign in southern Ethiopia: Community based cross–sectional study

Molalegn Mesele[1]*, Kelemu Abebe[1], Samuel Dessu[2], Walellign Anmut[1], Addisu Yeshambel[1], Zinabu Dawit[3], Tiwabwork Tekalign[4], Natnael Atnafu[1], Yohannes Fikadu[5]

1 School of Midwifery, College of Health Science and Medicine, Wolaita Sodo University, Wolaita Sodo, Ethiopia, 2 Department of Public Health, College of Medicine and Health Sciences, Wolkite University, Wolkite, Southern Ethiopia, 3 Department of Nursing, Arba Minch Health Science College, Arba Minch, Southern Ethiopia, 4 School of Nursing, College of Health Science and Medicine, Wolaita Sodo University, Wolaita Sodo, Ethiopia, 5 Department of Midwifery, College of Medicine and Health Sciences, Wolkite University, Wolkite, Southern Ethiopia

* emimolalegn@gmail.com

**Data Availability Statement:** All relevant data are within the paper and its Supporting information files.

## Abstract

### Background

Over the previous few decades, significant progress has been made in reducing newborn mortality, but the worldwide scale of the problem remains high. A considerable number of newborn death and difficulties owing to neonatal danger signs could be avoided if mothers sought appropriate health care for common neonatal risk indications, according to a number of studies presently underway in Ethiopia. The aim of this study is to assess health care seeking behavior of mothers' in related to neonatal danger signs.

### Method

A community-based cross-sectional study was conducted among 410 participants in Wolaita Sodo, From October 1 to October 30, 2019. To collect data, structured interviewer administered questionnaire was used. Data was coded, cleaned, recoded and entered in to epi-data version 3.1 and transported to SPSS window version 21 for analysis. Multivariable logistic regression was carried out and p-value of less than or equal to 0.05 was considered statistically significant.

### Result

A total of 410 mothers participated in this study, 110 (47.6%) mothers preferred health intuition for their neonate. Husband educational status (AOR = 2.4, 95% CI = 1.1, 5.5), communication media (AOR = 4.3, 95% CI = 2.4, 7.5), place of residence (AOR = 3.5, 95% C.I = 1.9, 6.7), ANC follow up (AOR = 2.8, 95% CI = 1.4, 5.8), and PNC follow (AOR = 1.7, 95% CI = 1.1, 3.1) were all factors that significantly associated with health care seeking practice neonatal dander signs.

**Funding:** The authors received no specific funding for this work.

**Competing interests:** The authors have declared that no competing interests exist.

## Conclusion

Overall, there was a low degree of health-seeking practice. The educational status of the mother's husband, communication media, residence, ANC follow-up, and PNC follow-up all predicted the mothers' health-care seeking behavior. The study also identifies the Wolaita Zone and Sodo town health offices, the health development army, one to five local community organizations with and health extension workers as key contributors.

## Introduction

A child's survival depends on the neonatal period, which lasts for the first 28 days of life [1]. When the infant is a newborn, its chances of survival are determined. To lessen mortality in children under the age of five, newborn mortality must be reduced [2]. To increase neonates' chances of survival and provide the groundwork for a healthy life, it is essential to give them with the right nourishment and care [3]. Preterm birth concerns, labor and delivery problems (complications connected to the intrapartum period), and infections are the main causes of newborn deaths worldwide. Three-quarters of all neonatal deaths can be attributed to these three causes when taken together [4]. By 2030, the Sustainable Development Goals (SDGs) seek to end the preventable deaths of infants and young children. All nations should work toward reducing the Neonatal Mortality Rate (NMR) to 12 deaths per 1,000 live births or less by 2030 and the under-5 mortality rate to 25 deaths per 1,000 live births or less [5].

In 2018, Sub-Saharan Africa had the highest NMR of death per 1,000 live births, which means that a child born in a low-income country is ten times more likely to die than a child born in a high-income country [6]. In Ethiopia neonatal mortality rate was 30 deaths per 1,000 live births in 2019; in 2016, Ethiopian Demographic and Health Survey, it was 29 [7, 8]. Evidences indicated that there is association between treatments seeking at health facilities and neonatal mortality [9, 10]. This indicates that, neonatal death is preventable when timely and appropriate care is given [11]. A study also indicated that one third of neonatal deaths can eliminated by caring for small and ill neonate. However, care seeking behavior for neonatal illness is low in low income and middle income countries [12].

Any action taken by people who know they have a health problem or are sick in order to obtain a proper treatment is referred to as health care seeking behavior. It entails recognizing symptoms, determining the nature of the sickness, and, most importantly, providing effective home care and monitoring. Knowledge of the illness's cause and treatment, as well as its length and perceived significance, socioeconomic status, and cultural customs, all influence health-care seeking behavior [13]. Millions of mothers and their neonate all over the world are living in a social environment that does not support health care seeking behavior [12]. Thus, many mothers did not generally seek formal health care during pregnancy, childbirth and puerpererium, which have a major impact on health care seeking for mothers and survival of their neonate [14] Neonatal danger sign refer to occurrence of sign that would show great danger of new born mortality and morbidity and necessity for initial therapeutic intervention. Convulsions, fever, lethargy, and poor breast milk feeding, as well as chest retractions, jaundice, and vomiting, are all significant danger signs that must be addressed immediately [15–18]. In Ethiopia, 40.7% of mothers reported having knowledge of newborn danger signs [19]

Ethiopia has made significant progress in the implementation of integrated health care, yet children continue to suffer from mortality and morbidities associated with danger indicators. This is mostly recognized to mother's health care seeking behavior [20–22]. In the southern

nation nationalities people region (SNNPR), a number of packages of interventions target newborn care. The health extension program is very important in delivering quality neonatal, maternal and child health services through effective and efficient linkages between health post, health center and community. Limited studies remained conducted in Ethiopia with respect to mothers' practice towards neonatal danger signs. So this research is intended to assess the mothers' health care seeking behavior towards neonatal danger signs and factors associated with seeking modern medical care for their sick neonates.

## Method and material

### Study setting and design

A community–based cross–sectional study was conducted on mothers who gave birth in the previous 12 months in Sodo town, Wolaita Zone, SNNPR, Ethiopia, from October 1 to October 30, 2019. Sodo town is 390 kilometers south of Ethiopia's capital, Addis Ababa, and 153 kilometers south of Ethiopia's regional capital, Hawasa. The town is divided into four administrative sub-cities. The town's overall population in 2018 was 182,607 people (93,130 men and 89,477 women), with 28,499 children under the age of five and 4576 infants under the age of one year, according to the town administration office. In the reproductive age range, there are also 4963 women (15–49 years). Functioning health facilities in the town includes 17 medium and lower level clinics, 17 health posts, 3 health centers and two Hospitals (one private and one government).

### Source and study population

All women in reproductive age group (15–49) living in Sodo town and two Sodo zurea kebele was source population and the potential study population from a list of those mothers who give birth in the last 12 months preceding the survey obtained from community health extension workers; each kebeles of the respective sub-city was study population.

### Sample size determination and sampling procedure

From a study conducted in Arba Minch General Hospital, sample size was estimated using the single population proportion formula, taking into account the following factors: marginal error of 0.05, 95 percent confidence interval, and p-value 41% [23]. By adding 10% non-response rate, the final sample size was 410.

Mothers who gave birth in the previous 12 months were coded by health extension workers, and the sample size was distributed proportionally to all administrative sub-cities of Sodo town and two Sodo zurea kebele, then every 6th woman was questioned using a systematic random sampling methodology.

### Measurement

Face-to-face interviews with structured questionnaires were used to collect data. The questionnaire, which includes socioeconomic and demographic characteristics, obstetric and practice of mother towards neonatal danger sign, the questionnaires were initially devised in English and then translated into Amharic. The mother and child health program of the Johns Hopkins Program for International Education in Gynecology and Obstetrics was used to prepare the questionnaire. Mothers' health-seeking activity in reaction to newborn danger signs is one of the study's findings [24].

## Variables

The dependent variable was health care seeking practice for neonatal danger signs, while the independent variables were socio-demographic and socio-economic features, obstetrics history, and the mother's preference seeking care for newborn danger sign.

## Operational definition

Care seeking practice: seeking medical or non-medical care in response to neonatal danger sign to reduce severity and complication after recognizing the danger signs and the perceived nature of illness and it was measured by [25].

Health care seeking practice for neonatal danger signs: those mothers who have taken their neonates to health facility immediately after the neonate has developed danger sign [26].

## Data processing and analysis

Data was coded, cleaned, recoded and entered in to epi-data version 3.1 and transported to SPSS window version 21 for analysis. Simple descriptive summary statistics was done. Table and statement was used to present the result of the data. Association between independent and dependent variables was analyzed first using bivariate logistic regression analysis. All variables with p-value less than 0.25 in bivariate logistic regression were entered to multivariable logistic regression for controlling possible confoundering and variable with p-value less than or equals to 0.05 considered as statistically significant.

## Ethics statement

The Wolaita Sodo University College of Health Science and Medicine's Ethical Review Board granted ethical approval.

Finally, a formal letter of permission from the Sodo Town Health Office was obtained in order to move forward with the data collecting. The study participants, as well as the parents or guardians of each participant under the age of 18, provided oral consent. Following that, participants' assent was requested, and those who were older than or equal to 18 years old gave their verbal approval. Participation was only permitted after receiving both assent and informed oral consent. To enable thorough and sincere self-disclosure, we kept voluntary engagement and confidentiality.

## Result

### Socio-demographic characteristics of participants

During the data collection process, 410 moms were interviewed, yielding a 100% response rate. The average age was 26.45 years (SD 6.2). In this study, the majority of mothers 173 (42.2%) were between the ages of 25 and 34, the majority of mothers had a primary (grades 1–8) educational level, and 407 (99.3%) were married (Table 1).

### Obstetrics history of participants

Regarding obstetric history of respondents; three 354 (86.3%) of respondents attend Antenatal Care (ANC) follows up for their last pregnancy. Two hundred twenty three (54.4%) have a history of Postnatal Care (PNC) services utilization (Table 2).

**Table 1. Socio demographic characteristics of mothers in Sodo town, Wolaita Zone, southern Ethiopia, 2019 (n = 410).**

| Variables | Frequency | Percentage (%) |
|---|---|---|
| Age of respondents | | |
| 18–24 | 169 | 41.2 |
| 25–34 | 173 | 42.2 |
| 34–49 and above | 68 | 16.6 |
| Marital status | | |
| Married | 407 | 99.3 |
| Divorced | 3 | 0.7 |
| Mothers educational level | | |
| Never attending school | 48 | 11.7 |
| primary education | 173 | 42.2 |
| secondary education | 152 | 37.1 |
| college and university | 37 | 9.0 |
| Husbands educational status | | |
| Never attending school | 44 | 10.7 |
| primary education | 87 | 21.2 |
| secondary education | 206 | 50.2 |
| college and university | 73 | 17.8 |
| Mothers occupational status | | |
| Merchant | 45 | 11.0 |
| Government employee | 35 | 8.5 |
| Housewife | 295 | 72.0 |
| Daily laborer | 20 | 4.9 |
| Students | 15 | 3.7 |
| Husband's occupation | | |
| Merchant | 109 | 26.6 |
| Government employee | 134 | 32.7 |
| Daily laborer | 71 | 17.3 |
| Students | 6 | 1.5 |
| Farmer | 90 | 21.9 |
| Types of communication media | | |
| Television | 132 | 32.2 |
| Radio | 278 | 67.8 |
| Residence | | |
| Urban | 286 | 69.8 |
| Rural | 124 | 30.2 |
| Family monthly income | | |
| 500-1000ETB | 288 | 70.2 |
| 1500-3000ETB | 71 | 17.3 |
| >3500ETB | 51 | 12.4 |

## Health care seeking practice of mothers for neonatal danger signs

231 (56.3%) mothers of all respondents said their newborns had encountered at least one of the WHO-recognized neonatal danger signs. In regarding of danger signs, the majority of neonates 98 (42.4%) had persistent vomiting, 64 (27.7% had convulsions, and 46 (20.1%) had breathing problems (Fig 1).

**Table 2. Obstetrics history of mothers in Sodo town, Wolaita Zone, Southern Ethiopia, 2019(n = 410).**

| Variables | Frequency (%) | Percentage (%) |
|---|---|---|
| Gravidity | | |
| Two | 157 | 38.3 |
| Three | 105 | 25.6 |
| >Three | 148 | 36.1 |
| Parity | | |
| One | 163 | 39.8 |
| Two | 104 | 25.4 |
| >Two | 143 | 34.9 |
| ANC follow up | | |
| Yes | 354 | 86.3 |
| No | 56 | 13.7 |
| PNC follow up | | |
| Yes | 323 | 78.8 |
| No | 87 | 21.2 |
| Place of delivery | | |
| Home | 7 | 1.7 |
| Health center | 71 | 17.3 |
| Hospital | 332 | 81.0 |
| Mode of delivery | | |
| Spontaneous vaginal delivery | 288 | 70.2 |
| Instrumental delivery | 110 | 26.8 |
| Ceserian section | 12 | 2.9 |

Mothers are not seeking medical care for a variety of reasons, including 54 (44.9%) expensive treatment costs, 22 (18.2%) minor illnesses, and 15 (12.4%) believing that home remedies are more effective (Table 3). The majority of mothers, 210 (51.2%), continued to breastfeed their ill neonates. The reasons for not continuing to breastfeed were 96 (48%) vomiting, 72 (36%) choke, and 22 (11%) diarrhea.

## Factors associated with maternal health care seeking practice about neonatal danger signs

Husband educational status, communication media, residence, ANC follow up, and PNC follow up were the factors that significantly affected maternal health care seeking practice for neonatal danger signals in multivariable logistic regression. The educational level of the husband is a significant factor in the mothers' health-care seeking practice. When compared to mothers' husband with no formal education, those with a husband's educational level of college and above were 2.4 times (AOR = 2.4, 95% CI = 1.1, 5.5) more likely to seek medical treatment for neonatal danger signs. Another factor that influences maternal health care seeking practice for neonatal danger signs is residency. Participants who lived in an urban area were more than 3.5 times (AOR = 3.5, 95% C.I = 1.9, 6.7) more likely than those who lived in a rural location to seek medical care for neonatal danger signs. Participants who had access to the media were more likely to seek medical care for newborn danger signs. When compared to participants who do not have access to mass media, those who do have access to mass media are 4.3 times (AOR = 4.3, 95% CI = 2.4, 7.5) more likely to seek medical treatment for newborn danger signals. Another factor that is strongly linked to mothers' health-care seeking practice

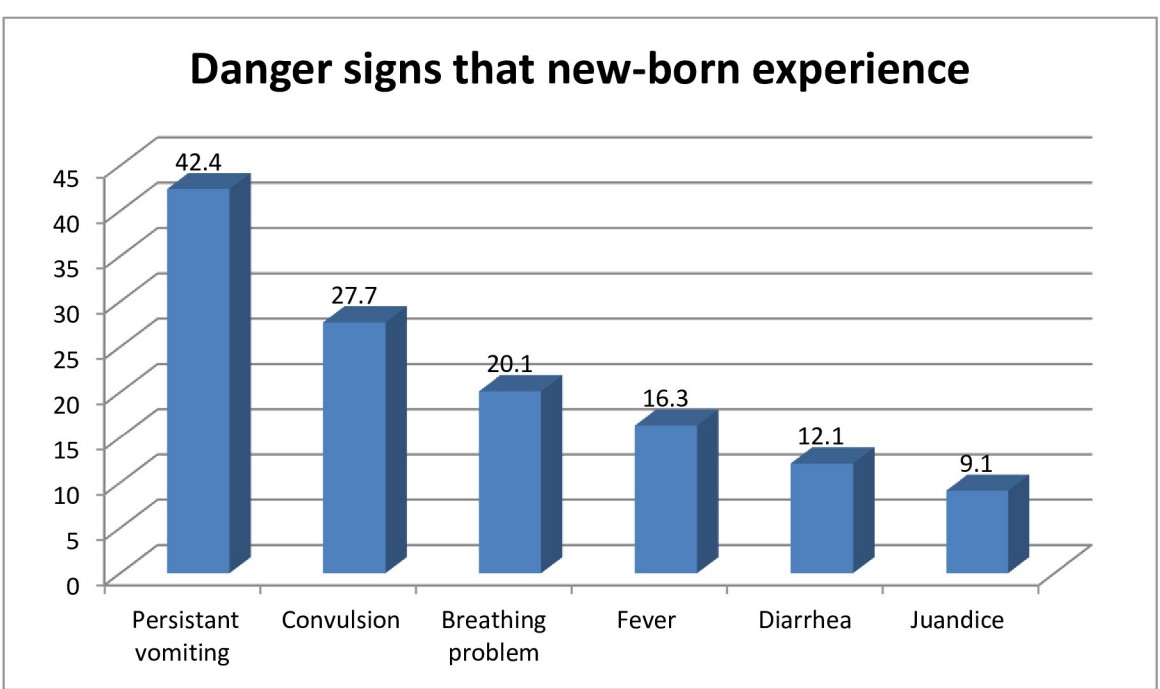

**Fig 1. Danger signs that new-born in experienced in Sodo town, Wolaita Zone, southern Ethiopia, 2019 (n = 231).** From a total of 231 neonates who experienced danger signs, 110 (47.6%) mothers preferred to seek care for their sick neonate at a health facility, 75 (32.4%) preferred traditional healers, and 18 (7.8%) gave home therapy (Fig 2). "Damakesie," garlic, "tenadam," honey, a mixture of lemon and ash put on the neonate's head for tonsillitis, match stick for convulsions, tepid sponging for fever, exposure to sunlight for jaundice, and rubbing with coconut oil for cold body were some of the home treatments mothers used for their sick children.

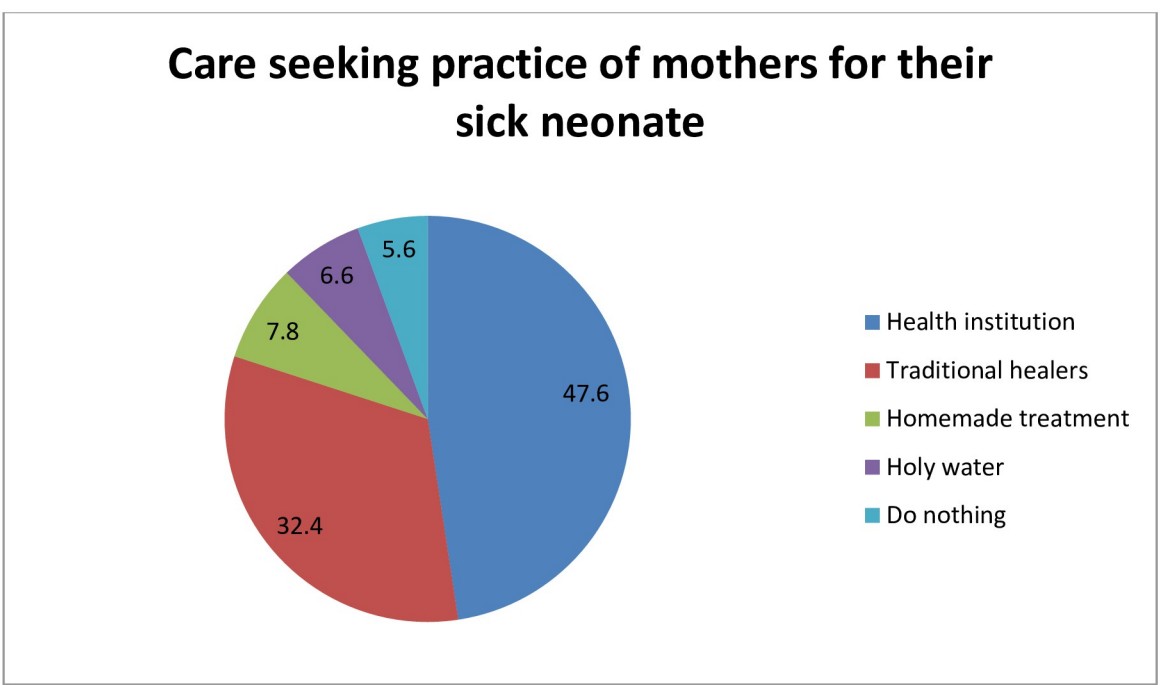

**Fig 2. Care seeking practice of mothers for their sick neonate in Sodo town, Wolaita Zone, southern Ethiopia, 2019 (n = 231).**

**Table 3. Reasons for not seeking medical care for neonatal danger signs in Sodo town, Wolaita Zone, southern Ethiopia, 2019 (n = 121).**

| Variable | Categories | Frequency | Percent (%) |
|---|---|---|---|
| Reasons for not seeking medical care | High treatment cost | 54 | 44.9 |
| | Illness was not serious | 22 | 18.2 |
| | Considering that home remedies are more effective | 15 | 12.4 |
| | Didn't trust facility/poor quality of care | 14 | 11.5 |
| | Lack of knowledge about danger signs | 13 | 10.7 |
| | No reason | 3 | 2.6 |

for newborn danger signs is ANC follow-up. When compared to mothers who had no history of ANC follow up, mothers who had ANC follow up were more than twice as likely to seek medical attention for neonatal danger signals (AOR = 2.8, 95% CI = 1.4, 5.8). Another characteristic that is strongly linked to mothers' health-care seeking behavior for newborn danger signs is PNC follow-up. When compared to mothers who had no history of PNC follow up, mothers who had PNC follow up were more than 1.7 times more likely to seek medical care for neonatal danger signals (AOR = 1.7, 95% CI = 1.1, 3.1) (Table 4).

## Discussion

The use of health-care seeking behaviors for neonatal danger signs holds great potential for lowering newborn mortality and morbidity. Improving mothers' health-seeking behavior for newborn danger signs can lower child morbidity and mortality; research suggest that delaying or refusing treatment contributes to a significant proportion of child mortality in underdeveloped nations [27, 28]. The current study indicates that 110 (47.6%) participants sought health care institution for their children during neonatal danger signs. The finding of the current

**Table 4. Factors associated with mothers health care seeking practice for neonatal danger signs in Sodo town, Wolaita Zone, southern Ethiopia, 2019 (n = 410).**

| Variables | Health care seeking practice | | Odds ratio with 95% CI |
|---|---|---|---|
| | Seeking medical care | Not seeking medical care | Adjusted |
| Husbands educational status | | | |
| No formal education | 25(56.8%) | 19(43.2%) | 1 |
| Primary education | 41(47.1%) | 46(52.9%) | 1.8(0.7,4.9) |
| Secondary education | 107(51.7%) | 99(48.1%) | 2.9(0.8,4.8) |
| College and above | 56(76.7%) | 17(23.3%) | 2.4(1.1,5.5)* |
| Residence | | | |
| Urban | 171(59.8%) | 115(40.2%) | 3.5(1.9,6.7)* |
| Rural | 58(46.8%) | 66(53.2%) | 1 |
| Communication media | | | |
| Television | 103(72%) | 40(26%) | |
| Radio | 126(47.2%) | 141(52.8%) | 4.3(2.4,7.5) |
| ANC follow up | | | |
| Yes | 211(59.6%) | 143(40.4%) | 2.8(1.4,5.8)* |
| No | 18(32.1%) | 38(67.9%) | 1 |
| PNC follow up | | | |
| Yes | 195(60.4%) | 128(39.6%) | 1.7(1.1,3.1)* |
| No | 34(39.1%) | 53(60.9%) | 1 |

*Significant association at P-value <0.05

study is in line with a study done in North West Ethiopia (48.8%) [29], Enugu state, Nigeria (47.7%) [30], but higher as compared with a study done in Wolkite town, Gurage Zone (32%) [31], northern India (23%) [32], and urban slum of India (18.1%) [33]. The finding of the current study is lower as compared with a study done in Bahrdar 82.7% [34]. The difference might be due to the time and socio-economic and socio-demographic difference.

One of the factors that was strongly linked to the mother's health-care seeking behavior for newborn danger signs was her husband's educational status. Those who had a husband with a college education or above were more likely to seek medical treatment for neonatal danger signs than those who had a husband with no formal education. The current study's findings supported by a study conducted in Wolkite Town [31]. The reason for this could be that husbands with a higher level of education (college and above) have a better chance of accessing knowledge and comprehending the benefits of seeking medical care because husbands are the head of the house, the decision maker, and the primary source of income in most families. The educational status of the husband has influenced the mother's health care seeking practice for neonatal danger signs.

Residency is another element that influences maternal health care seeking for newborn danger signs. Participants who lived in an urban area were more likely to seek medical care for newborn danger signs than those who resided in a rural area. A study conducted in Ethiopia's Derra District, North Shoa Zone, Oromia Regional State, found that urban inhabitants are more likely than rural ones to seek medical treatment for neonatal danger signs [20].

Participants with media access were more likely to seek medical attention for neonatal danger signs, when compared to individuals who did not have access to the media, those who did have a higher likelihood of seeking medical attention for newborn danger signals. The possible reason might be those who have mass media access information that helps the mother to medical care for their sick neonate.

ANC follow-up is another characteristic that is highly associated with mothers' health-care seeking practice for infant danger signs. ANC follow-up mothers were more likely to seek medical assistance for newborn danger signs than mothers who had no history of ANC follow-up. A study conducted in Fiche town, Oromia region, Ethiopia, supported with the findings of the current study [35]. PNC follow-up is another factor that is closely associated to mothers' health-care seeking behavior for newborn danger signs. PNC follow-up mothers were more likely to seek medical care for neonatal danger signals than moms who had no history of PNC follow-up. Several investigations, including Fiche town in Ethiopia's Oromia region, supported with the current study's findings [35], Wolkite town in the Gurage Zone in southern Ethiopia [30] and Ambo town in central Ethiopia [25]. The likely reason is that at PNC follow-ups, a mother may receive newborn care counseling before discharge, which may enhance the mother's health-seeking behavior for neonatal risk indicators.

The high sample size, well-trained research assistants who interviewed the participants, and the community-based study were all strengths of this study. Because the study's cross-sectional design makes determining a temporal relationship between the dependent and independent variables difficult, a prospective follow-up study can provide more information than a cross-sectional study.

## Conclusion

The percentage of mothers who sought health care for neonatal danger signs was relatively low. The factors that substantially affected mothers' health care seeking practice for neonates were husband educational status, communication media, residence, ANC follow up, and PNC follows up. The current study's findings point to the necessity for intervention focused on

increasing mother health-care seeking behavior for common neonatal danger signs. The findings additionally implicate the Wolaita Zone and Sodo town health officials, the health development army, and one to five local community groups as major players. This could be accomplished through a variety of community-based platforms, such as gatherings, home visits, and existing community service.

## Supporting information

**S1 File.**
(DOCX)

**S1 Dataset. Underlying data set.**
(SAV)

## Acknowledgments

We would like to express our heartfelt thanks for Wolaita Sodo University, all individual for their contribution and cooperation.

## Author Contributions

**Conceptualization:** Molalegn Mesele, Addisu Yeshambel, Natnael Atnafu.

**Data curation:** Molalegn Mesele.

**Formal analysis:** Molalegn Mesele, Walellign Anmut.

**Funding acquisition:** Molalegn Mesele.

**Investigation:** Molalegn Mesele.

**Methodology:** Molalegn Mesele.

**Project administration:** Molalegn Mesele.

**Resources:** Molalegn Mesele.

**Software:** Molalegn Mesele, Samuel Dessu.

**Supervision:** Molalegn Mesele, Kelemu Abebe, Samuel Dessu, Walellign Anmut, Addisu Yeshambel, Zinabu Dawit, Tiwabwork Tekalign, Natnael Atnafu, Yohannes Fikadu.

**Validation:** Molalegn Mesele.

**Visualization:** Molalegn Mesele.

**Writing – original draft:** Molalegn Mesele, Addisu Yeshambel, Natnael Atnafu.

**Writing – review & editing:** Molalegn Mesele, Kelemu Abebe, Samuel Dessu, Walellign Anmut, Addisu Yeshambel, Zinabu Dawit, Tiwabwork Tekalign, Natnael Atnafu, Yohannes Fikadu.

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
