## [Decision Letter · Decision Letter 0]

1 Apr 2022

PONE-D-21-24134

Mothers’ Health Care Seeking Practice for Neonatal Danger Sign in Southern Ethiopia: Community Based Cross–Sectional Study

PLOS ONE

Dear Dr. Mesele,

Thank you for submitting your manuscript to PLOS ONE. After careful consideration, we feel that it has merit but does not fully meet PLOS ONE’s publication criteria as it currently stands. Therefore, we invite you to submit a revised version of the manuscript that addresses the points raised during the review process.

We look forward to receiving your revised manuscript.

Kind regards,

Sungwoo Lim, DrPH

Academic Editor

PLOS ONE

a) Did participants provide their written or verbal informed consent to participate in this study?

3. You indicated that you had ethical approval for your study. In your Methods section, please ensure you have also stated whether you obtained consent from parents or guardians of the minors included in the study or whether the research ethics committee or IRB specifically waived the need for their consent.

4. Please include additional information regarding the survey or questionnaire used in the study and ensure that you have provided sufficient details that others could replicate the analyses. For instance, if you developed a questionnaire as part of this study and it is not under a copyright more restrictive than CC-BY, please include a copy, in both the original language and English, as Supporting Information.

“This study did not obtain any particular funding from government, commercial, or non-profit funding agencies.”

d) If you did not receive any funding for this study, please state: “The authors received no specific funding for this work.

6. Thank you for stating the following in your Competing Interests section: 

“There is no competing interest”

7. In your Data Availability statement, you have not specified where the minimal data set underlying the results described in your manuscript can be found. PLOS defines a study's minimal data set as the underlying data used to reach the conclusions drawn in the manuscript and any additional data required to replicate the reported study findings in their entirety. All PLOS journals require that the minimal data set be made fully available. For more information about our data policy, please see http://journals.plos.org/plosone/s/data-availability.

Reviewers' comments:

Reviewer's Responses to Questions

**Comments to the Author**

1. Is the manuscript technically sound, and do the data support the conclusions?

Reviewer #1: Partly

Reviewer #2: Partly

2. Has the statistical analysis been performed appropriately and rigorously? 

Reviewer #1: No

Reviewer #2: No

3. Have the authors made all data underlying the findings in their manuscript fully available?

Reviewer #1: Yes

Reviewer #2: Yes

4. Is the manuscript presented in an intelligible fashion and written in standard English?

Reviewer #1: No

Reviewer #2: No

5. Review Comments to the Author

Reviewer #1: TITLE: edit to read maternal health seeking behavior.....

INTRODUCTION: focus your literature on WHO danger signs. which danger signs were studied and why? specify and justify.Wat guidelines are used in Ethiopia for neonatal care? how does a mother learn about these signs, if they don't attend anc/pnc? are there community health workers in Ethiopia?

METHODS: is it a retrospective study? how do you deal with recall bias? indicate ethical approval number.

RESULTS: ensure all percentages add to 100%, seen many errors.summarize the tables- combine column 2 and 3 eg. 169(41.2%). quantify herbal use meds. how is husband s education important in this study?

DISSCUSSION: discuss the results and don't repeat results here. requires to be re written focusing on neonate, danger signs and mother. which danger sign was highly missed and why? this will help to implement better guidelines and health seeking behaviour.

Reviewer #2: Review for:

Mothers’ Health Care Seeking Practice for Neonatal Danger Sign in Southern Ethiopia: Community Based Cross–Sectional Study

General comment:

This is an important topic that the study tries to investigate, and a community based study that potentially includes community members that might could easily be excluded from healthcare by their social or economic status. However, there could still be some improvements to the study. An assessment of the knowledge, and the attitudes in addition to the practice would have added a significant lot of value to validate the results. And to assess the practice, the mothers actually needed to have a knowledge of the danger of these danger signs. The factors associated with health care seeking practice identified in the study such as husband’s education status, access to media, urban residence, ANC follow ups etc, clearly point towards access ‘Knowledge’.

The writing could also be improved with the help of a native English editor.

Specific comments:

Abstract:

Online 34, the authors need to indicate what statistical tests were carried out before the sentence “A p-value of less than or equal to 0.05 in a multivariable……..”

Introduction:

On pg 10, line 91, the sentence that starts with “Solitary limited studies remained conducted in Ethiopia with respect to mothers’ practice towards neonatal danger signs” should be referenced. The findings in this solitary study should be mentioned and other studies in the region could be reviewed as well. The gaps in those other studies should then be outlined that this particular study promises to close. The real scientific addition from the study should then be articulated after this sentence: “ So this research is intended to assess the ……” lines 92-94

Methods sections

If the authors insist on only assessing the practice of health seeking for danger signs, then the study population should only include those mothers who recognized danger signs in their babies, and the sample size calculated should only be for these population. There is no need to include the mothers whose babies had no danger signs in the initial interviews and in the analysis.

The sampling procedure on Pg 11, lines 116-19 is not clear, how the coding was done and every 6th woman is selected- these need to be explained in more detail.

Results

As already stated above, the effective sample size in this study so far is 231 NOT 410. If it is logistically difficult to get more participants into the study, perhaps the authors should acknowledge this as a limitation in the discussion sections.

Is it possible that care seeking practice could also vary by the particular danger sign, it would be nice to see if there any particular signs for which mothers would most likely seek for care compared to others?

Discussions

Please include study limitations in the discussions as they could be several, add the strengths of the study as well.

6. PLOS authors have the option to publish the peer review history of their article (what does this mean?). If published, this will include your full peer review and any attached files.

Reviewer #1: No

Reviewer #2: **Yes: **Pontius Bayo

---

## [Author Response · Author response to Decision Letter 0]

11 Aug 2022

Response letter 

Dear editors and reviewers:

Thank you for considering my manuscript for publication in your reputable journal. I have noted that your comments are valuable and make my research article well improved. I have made appropriate changes in the main document. Here in below are the point by point responses for your concern.

Academic Editor Comments to the Author

 Author’s response: the manuscript revised and written with PLOS ONE's style

a) Did participants provide their written or verbal informed consent to participate in this study?

 Author’s response: participants get verbal informed consent to participate in this study. I have no special reason to give verbal informed consent to participants. It is well explained on ethical approval and consent to participate section. During data collection respondents were requested their willingness to participate; those who were willing to participate were included in this study, but those who were not willing to participate were excluded from the study. Ethical approval was obtained from Wolaita Sodo University College of health science and medicine Ethical review board

3. You indicated that you had ethical approval for your study. In your Methods section, please ensure you have also stated whether you obtained consent from parents or guardians of the minors included in the study or whether the research ethics committee or IRB specifically waived the need for their consent.

 Author’s response: Oral informed consent was obtained from the parents of the minors

4. Please include additional information regarding the survey or questionnaire used in the study and ensure that you have provided sufficient details that others could replicate the analyses. For instance, if you developed a questionnaire as part of this study and it is not under a copyright more restrictive than CC-BY, please include a copy, in both the original language and English, as Supporting Information. 

 Author’s response: I will include a copy of questionnaire as Supporting Information. 

5. . Thank you for stating the following financial disclosure:

“This study did not obtain any particular funding from govern.ent, commercial, or non-profit funding agencies.”

d) If you did not receive any funding for this study, please state: “The authors received no specific funding for this work.

 Author’s response: This study did not obtain any particular funding from government, commercial, or non-profit funding agencies

6. Thank you for stating the following in your Competing Interests section: 

“There is no competing interest”

 Author’s response: I will complete your Competing Interests on the online submission

Reviewer Comments to the Author

1. edit title to read maternal health seeking behavior

 Author’s response: the title edited as health seeking behavior

2. focus your literature on WHO danger signs. Which danger signs were studied and why? specify and justify.Wat guidelines are used in Ethiopia for neonatal care? how does a mother learn about these signs, if they don't attend anc/pnc? are there community health workers in Ethiopia?

 Author’s response: Convulsions, fever, lethargy, and poor breast milk feeding, as well as chest retractions, jaundice, and vomiting are neonatal danger signs this study focus. Most of the time children death because of the above listed neonatal danger signs. INTEGRATED MANAGEMENT OF NEWBORN AND CHILDHOOD ILLNESS and guidelines are used in Ethiopia for neonatal care. Health extension workers play a great role in Ethiopia to teach mothers about the neonatal danger signs and mothers who have ANC/PNC follow up get information from health care providers. 

3. METHODS: is it a retrospective study? how do you deal with recall bias? indicate ethical approval number.

 Author’s response: data was collected from mothers who give birth in the last 12 months preceding the survey so I have not face with recall bias. Mothers mostly didn’t forget what happened on their children. ethical approval reference number is CHSM/ERC/08 

4. RESULTS: ensure all percentages add to 100%, seen many errors.summarize the tables- combine column 2 and 3 eg. 169(41.2%). quantify herbal use meds. how is husband s education important in this study?

 Author’s response: all percentages were added to 100%. In Ethiopia husbands have a great influence on all aspects of their wife. 

5. Discuss the results and don't repeat results here. requires to be re written focusing on neonate, danger signs and mother. which danger sign was highly missed and why? this will help to implement better guidelines and health seeking behavior.

 Author’s response: the discussion part have revised 

6. An assessment of the knowledge, and the attitudes in addition to the practice would have added a significant lot of value to validate the results. And to assess the practice, the mothers actually needed to have a knowledge of the danger of these danger signs. The factors associated with health care seeking practice identified in the study such as husband’s education status, access to media, urban residence, ANC follow ups etc, clearly point towards access ‘Knowledge’.

The writing could also be improved with the help of a native English editor.

 Author’s response: prior to assess the practice or behavior the knowledge and the practice part have assessed separately. The manuscript had revised with native English editor.

7. Online 34, the authors need to indicate what statistical tests were carried out before the sentence “A p-value of less than or equal to 0.05 in a multivariable……..”

 Author’s response: on abstract, before writing about p-value the statistical test indicated.

8. On pg 10, line 91, the sentence that starts with “Solitary limited studies remained conducted in Ethiopia with respect to mothers’ practice towards neonatal danger signs” should be referenced. The findings in this solitary study should be mentioned and other studies in the region could be reviewed as well. The gaps in those other studies should then be outlined that this particular study promises to close. The real scientific addition from the study should then be articulated after this sentence: “ So this research is intended to assess the ……” lines 92-94

 Author’s response: the term used ‘Solitary’ is not to stat only but to explain as little or no enough study have conducted in Ethiopia. Please understand what I want to say. 

9. If the authors insist on only assessing the practice of health seeking for danger signs, then the study population should only include those mothers who recognized danger signs in their babies, and the sample size calculated should only be for these population. There is no need to include the mothers whose babies had no danger signs in the initial interviews and in the analysis.

The sampling procedure on Pg 11, lines 116-19 is not clear, how the coding was done and every 6th woman is selected- these need to be explained in more detail. As already stated above, the effective sample size in this study so far is 231 NOT 410. If it is logistically difficult to get more participants into the study, perhaps the authors should acknowledge this as a limitation in the discussion sections.

 Author’s response: even if the authors intention is to see health care practice of mother for neonatal danger sign, from 410 we get 231(56.3%) of mothers had encountered at least one of the WHO-recognized neonatal danger signs. So we observe the prevalence of neonatal danger sign which is 56.3% prior talking about practice of mothers who face neonatal danger sign. 

Response letter 

Dear editors and reviewers:

Thank you for considering my manuscript for publication in your reputable journal. I have noted that your comments are valuable and make my research article well improved. I have made appropriate changes in the main document. Here in below are the point by point responses for your concern.

Academic Editor Comments to the Author

 Author’s response: the manuscript revised and written with PLOS ONE's style

a) Did participants provide their written or verbal informed consent to participate in this study?

 Author’s response: participants get verbal informed consent to participate in this study. I have no special reason to give verbal informed consent to participants. It is well explained on ethical approval and consent to participate section. During data collection respondents were requested their willingness to participate; those who were willing to participate were included in this study, but those who were not willing to participate were excluded from the study. Ethical approval was obtained from Wolaita Sodo University College of health science and medicine Ethical review board

3. You indicated that you had ethical approval for your study. In your Methods section, please ensure you have also stated whether you obtained consent from parents or guardians of the minors included in the study or whether the research ethics committee or IRB specifically waived the need for their consent.

 Author’s response: Oral informed consent was obtained from the parents of the minors

4. Please include additional information regarding the survey or questionnaire used in the study and ensure that you have provided sufficient details that others could replicate the analyses. For instance, if you developed a questionnaire as part of this study and it is not under a copyright more restrictive than CC-BY, please include a copy, in both the original language and English, as Supporting Information. 

 Author’s response: I will include a copy of questionnaire as Supporting Information. 

5. . Thank you for stating the following financial disclosure:

“This study did not obtain any particular funding from govern.ent, commercial, or non-profit funding agencies.”

d) If you did not receive any funding for this study, please state: “The authors received no specific funding for this work.

 Author’s response: This study did not obtain any particular funding from government, commercial, or non-profit funding agencies

6. Thank you for stating the following in your Competing Interests section: 

“There is no competing interest”

 Author’s response: I will complete your Competing Interests on the online submission

Reviewer Comments to the Author

1. edit title to read maternal health seeking behavior

 Author’s response: the title edited as health seeking behavior

2. focus your literature on WHO danger signs. Which danger signs were studied and why? specify and justify.Wat guidelines are used in Ethiopia for neonatal care? how does a mother learn about these signs, if they don't attend anc/pnc? are there community health workers in Ethiopia?

 Author’s response: Convulsions, fever, lethargy, and poor breast milk feeding, as well as chest retractions, jaundice, and vomiting are neonatal danger signs this study focus. Most of the time children death because of the above listed neonatal danger signs. INTEGRATED MANAGEMENT OF NEWBORN AND CHILDHOOD ILLNESS and guidelines are used in Ethiopia for neonatal care. Health extension workers play a great role in Ethiopia to teach mothers about the neonatal danger signs and mothers who have ANC/PNC follow up get information from health care providers. 

3. METHODS: is it a retrospective study? how do you deal with recall bias? indicate ethical approval number.

 Author’s response: data was collected from mothers who give birth in the last 12 months preceding the survey so I have not face with recall bias. Mothers mostly didn’t forget what happened on their children. ethical approval reference number is CHSM/ERC/08 

4. RESULTS: ensure all percentages add to 100%, seen many errors.summarize the tables- combine column 2 and 3 eg. 169(41.2%). quantify herbal use meds. how is husband s education important in this study?

 Author’s response: all percentages were added to 100%. In Ethiopia husbands have a great influence on all aspects of their wife. 

5. Discuss the results and don't repeat results here. requires to be re written focusing on neonate, danger signs and mother. which danger sign was highly missed and why? this will help to implement better guidelines and health seeking behavior.

 Author’s response: the discussion part have revised 

6. An assessment of the knowledge, and the attitudes in addition to the practice would have added a significant lot of value to validate the results. And to assess the practice, the mothers actually needed to have a knowledge of the danger of these danger signs. The factors associated with health care seeking practice identified in the study such as husband’s education status, access to media, urban residence, ANC follow ups etc, clearly point towards access ‘Knowledge’.

The writing could also be improved with the help of a native English editor.

 Author’s response: prior to assess the practice or behavior the knowledge and the practice part have assessed separately. The manuscript had revised with native English editor.

7. Online 34, the authors need to indicate what statistical tests were carried out before the sentence “A p-value of less than or equal to 0.05 in a multivariable……..”

 Author’s response: on abstract, before writing about p-value the statistical test indicated.

8. On pg 10, line 91, the sentence that starts with “Solitary limited studies remained conducted in Ethiopia with respect to mothers’ practice towards neonatal danger signs” should be referenced. The findings in this solitary study should be mentioned and other studies in the region could be reviewed as well. The gaps in those other studies should then be outlined that this particular study promises to close. The real scientific addition from the study should then be articulated after this sentence: “ So this research is intended to assess the ……” lines 92-94

 Author’s response: the term used ‘Solitary’ is not to stat only but to explain as little or no enough study have conducted in Ethiopia. Please understand what I want to say. 

9. If the authors insist on only assessing the practice of health seeking for danger signs, then the study population should only include those mothers who recognized danger signs in their babies, and the sample size calculated should only be for these population. There is no need to include the mothers whose babies had no danger signs in the initial interviews and in the analysis.

The sampling procedure on Pg 11, lines 116-19 is not clear, how the coding was done and every 6th woman is selected- these need to be explained in more detail. As already stated above, the effective sample size in this study so far is 231 NOT 410. If it is logistically difficult to get more participants into the study, perhaps the authors should acknowledge this as a limitation in the discussion sections.

 Author’s response: even if the authors intention is to see health care practice of mother for neonatal danger sign, from 410 we get 231(56.3%) of mothers had encountered at least one of the WHO-recognized neonatal danger signs. So we observe the prevalence of neonatal danger sign which is 56.3% prior talking about practice of mothers who face neonatal danger sign.

---

## [Decision Letter · Decision Letter 1]

19 Sep 2022

PONE-D-21-24134R1Mothers’ Health Care Seeking Behavior for Neonatal Danger Sign in Southern Ethiopia: Community Based Cross–Sectional StudyPLOS ONE

Dear Dr. Mesele,

Thank you for submitting your manuscript to PLOS ONE. After careful consideration, we feel that it has merit but does not fully meet PLOS ONE’s publication criteria as it currently stands. Therefore, we invite you to submit a revised version of the manuscript that addresses the points raised during the review process.

We look forward to receiving your revised manuscript.

Kind regards,

Sungwoo Lim, DrPH

Academic Editor

PLOS ONE

Reviewers' comments:

Reviewer's Responses to Questions

**Comments to the Author**

1. If the authors have adequately addressed your comments raised in a previous round of review and you feel that this manuscript is now acceptable for publication, you may indicate that here to bypass the “Comments to the Author” section, enter your conflict of interest statement in the “Confidential to Editor” section, and submit your "Accept" recommendation.

Reviewer #1: All comments have been addressed

Reviewer #2: (No Response)

2. Is the manuscript technically sound, and do the data support the conclusions?

Reviewer #1: Yes

Reviewer #2: No

3. Has the statistical analysis been performed appropriately and rigorously? 

Reviewer #1: Yes

Reviewer #2: No

4. Have the authors made all data underlying the findings in their manuscript fully available?

Reviewer #1: Yes

Reviewer #2: Yes

5. Is the manuscript presented in an intelligible fashion and written in standard English?

Reviewer #1: Yes

Reviewer #2: No

6. Review Comments to the Author

Reviewer #1: this is an important study , the corrections ahve been done adequately and the manuscript can be accepted now.

Reviewer #2: Comments:

1. While the subject matter in this study is important, the grammar still needs a lot of improvement at least to show some of the missing references and an understanding of definitions. For example in introduction: The first sentence line 54 requires an addition of a reference. The sentence is also technically confusing. Which is the newborn period? As far as I know, this period is up to one month while the infants are up to one year.

2. Many comments are not yet adequately addressed:

a. The authors acknowledge on line 77 pg13 that the ‘knowledge’ of danger signs influence the ‘practice’ of health seeking and their response is simply that this has been assessed separately. I suggest the authors include this in their literature review under introduction and provide the knowledge levels that currently exist.

b. The response “the term used ‘Solitary’ is not to stat only but to explain as little or no enough study have conducted in Ethiopia. Please understand what I want to say” is still not adequate in my view. I see the word ‘solitary’ has been changed to ‘limited’ but the issue raised in the comment was actually to discuss the findings of these limited studies and identify the gaps that this study would like to address. This is usually the origin of new studies covering the same subject.

c. In methods section: The issue of sampling and sample size remains largely unaddressed. Yes, the prevalence of danger signs is 56.3% but this is not what the study is set up for. It is the those who developed these danger signs we are interested in- how many sought care and what influenced their behavior?

7. PLOS authors have the option to publish the peer review history of their article (what does this mean?). If published, this will include your full peer review and any attached files.

Reviewer #1: **Yes: **Dr Varsha Vekaria-Hirani

Reviewer #2: No

---

## [Author Response · Author response to Decision Letter 1]

6 Oct 2022

Response letter 

Dear editors and reviewers:

Thank you for considering my manuscript for publication in your reputable journal. I have noted that your comments are valuable and make my research article well improved. I have made appropriate changes in the main document. Here in below are the point by point responses for your concern.

Reviewer Comments to the Author

1. The first sentence line 54 requires an addition of a reference. The sentence is also technically confusing. Which is the newborn period? As far as I know, this period is up to one month while the infants are up to one year.

 Author’s response: The sentence and the reference have corrected. As WHO define, ‘A newborn infant, or neonate, is a child under 28 days of age’. The reviewers saying as it is written in revised version.

2. The authors acknowledge on line 77 pg13 that the ‘knowledge’ of danger signs influence the ‘practice’ of health seeking and their response is simply that this has been assessed separately. I suggest the authors include this in their literature review under introduction and provide the knowledge levels that currently exist

 Author’s response: The aim of this research is to assess health care seeking behavior of mothers’ in related to neonatal danger signs. Knowledge level is not the aim of this study and it is possible to assess separately. On line 77 it says “Knowledge of the illness's cause and treatment” influence health-care seeking behavior 

3. The response “the term used ‘Solitary’ is not to stat only but to explain as little or no enough study have conducted in Ethiopia. Please understand what I want to say” is still not adequate in my view. I see the word ‘solitary’ has been changed to ‘limited’ but the issue raised in the comment was actually to discuss the findings of these limited studies and identify the gaps that this study would like to address. This is usually the origin of new studies covering the same subject.

Author’s response: dear reviewer I understand what you want to say. Some research has done in Ethiopia but it is not enough regarding this very critical area because most women loss their children due to neonatal danger sign. 

4. In methods section: The issue of sampling and sample size remains largely unaddressed. Yes, the prevalence of danger signs is 56.3% but this is not what the study is set up for. It is the those who developed these danger signs we are interested in- how many sought care and what influenced their behavior?

 Author’s response: numbers of mothers seeking care from health institution and factors influenced their behavior have stated in the manuscript

---

## [Editor Report · Decision Letter 2]

23 Nov 2022

PONE-D-21-24134R2Mothers’ Health Care Seeking Behavior for Neonatal Danger Sign in Southern Ethiopia: Community Based Cross–Sectional Study

PLOS ONE

Dear Dr. Mesele,

Thank you for submitting your manuscript to PLOS ONE. After careful consideration, we feel that it has merit but does not fully meet PLOS ONE’s publication criteria as it currently stands. Therefore, we invite you to submit a revised version of the manuscript that addresses the points raised during the review process.

We look forward to receiving your revised manuscript.

Kind regards,

Sungwoo Lim, DrPH

Academic Editor

PLOS ONE

Journal Requirements:

Additional Editor Comments (if provided):

We think that current responses and edits do not sufficiently address comments raised by reviewers. Please make sure to adequately address reviewers' comments (attached below) with full details and revisions. 

Comments:

1. While the subject matter in this study is important, the grammar still needs a lot of improvement at least to show some of the missing references and an understanding of definitions. For example in introduction: The first sentence line 54 requires an addition of a reference. The sentence is also technically confusing. Which is the newborn period? As far as I know, this period is up to one month while the infants are up to one year.

2. Many comments are not yet adequately addressed:

a. The authors acknowledge on line 77 pg13 that the ‘knowledge’ of danger signs influence the ‘practice’ of health seeking and their response is simply that this has been assessed separately. I suggest the authors include this in their literature review under introduction and provide the knowledge levels that currently exist.

b. The response “the term used ‘Solitary’ is not to stat only but to explain as little or no enough study have conducted in Ethiopia. Please understand what I want to say” is still not adequate in my view. I see the word ‘solitary’ has been changed to ‘limited’ but the issue raised in the comment was actually to discuss the findings of these limited studies and identify the gaps that this study would like to address. This is usually the origin of new studies covering the same subject.

c. In methods section: The issue of sampling and sample size remains largely unaddressed. Yes, the prevalence of danger signs is 56.3% but this is not what the study is set up for. It is the those who developed these danger signs we are interested in- how many sought care and what influenced their behavior?
---

## [Author Response · Author response to Decision Letter 2]

6 Jan 2023

Response letter 

Dear editors and reviewers:

Thank you for considering my manuscript for publication in your reputable journal. I have noted that your comments are valuable and make my research article well improved. I have made appropriate changes in the main document. Here in below are the point by point responses for your concern.

Reviewer Comments to the Author

1. The first sentence line 54 requires an addition of a reference. The sentence is also technically confusing. Which is the newborn period? As far as I know, this period is up to one month while the infants are up to one year.

 Author’s response: The sentence and the reference have corrected. As WHO define, ‘A newborn infant, or neonate, is a child under 28 days of age’. The reviewers saying as it is written in revised version.

2. The authors acknowledge on line 77 pg13 that the ‘knowledge’ of danger signs influence the ‘practice’ of health seeking and their response is simply that this has been assessed separately. I suggest the authors include this in their literature review under introduction and provide the knowledge levels that currently exist

 Author’s response: the current knowledge level have been incorporated under introduction from a systematic review conducted in Ethiopia 

3. The response “the term used ‘Solitary’ is not to stat only but to explain as little or no enough study have conducted in Ethiopia. Please understand what I want to say” is still not adequate in my view. I see the word ‘solitary’ has been changed to ‘limited’ but the issue raised in the comment was actually to discuss the findings of these limited studies and identify the gaps that this study would like to address. This is usually the origin of new studies covering the same subject.

Author’s response: dear reviewer I understand what you want to say. Some research has done in Ethiopia but it is not enough regarding this very critical area because most women loss their children due to neonatal danger sign. 

4. In methods section: The issue of sampling and sample size remains largely unaddressed. Yes, the prevalence of danger signs is 56.3% but this is not what the study is set up for. It is the those who developed these danger signs we are interested in- how many sought care and what influenced their behavior?

 Author’s response: numbers of mothers seeking care from health institution and factors influenced their behavior have stated in the manuscript

---

## [Editor Report · Decision Letter 3]

13 Jan 2023

Mothers’ Health Care Seeking Behavior for Neonatal Danger Sign in Southern Ethiopia: Community Based Cross–Sectional Study

PONE-D-21-24134R3

Dear Dr. Mesele,

We’re pleased to inform you that your manuscript has been judged scientifically suitable for publication and will be formally accepted for publication once it meets all outstanding technical requirements.

Kind regards,

Sungwoo Lim, DrPH

Academic Editor

PLOS ONE

Additional Editor Comments (optional):

All the comments have been adequately addressed.  
---

## [Editor Report · Acceptance letter]

30 Mar 2023

PONE-D-21-24134R3 

Mothers’ Health Care Seeking Behavior for Neonatal Danger Sign in Southern Ethiopia: Community Based Cross–Sectional Study 

Dear Dr. Mesele:

I'm pleased to inform you that your manuscript has been deemed suitable for publication in PLOS ONE. Congratulations! Your manuscript is now with our production department. 

Kind regards, 

on behalf of

Dr. Sungwoo Lim 

Academic Editor

PLOS ONE